# Towards Money Market in General Equilibrium Framework

Truong Hong Trinh

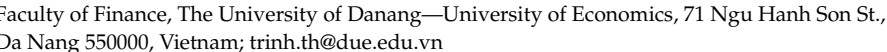

Faculty of Finance, The University of Danang—University of Economics, 71 Ngu Hanh Son St.,
Da Nang 550000, Vietnam; trinh.th@due.edu.vn

**Abstract:** This paper aims to integrate the money market into the structure of the economy. The microfoundation is the starting point to define the money market and the general equilibrium mechanism of the economy. On this basis, this research seeks a linking mechanism of the money market with economic activity in the general equilibrium framework. The relationships between money supply and national outcome, inflation, and price level are studied in three cases: full-employment equilibrium economy, steady-state equilibrium economy, and sticky-price equilibrium economy. The research result explains the interrelation and transmission mechanism between the money market and the general equilibrium of the economy. The paper provides the theoretical foundation for further research on the money market and monetary policies towards economic growth and macroeconomic stability.

**Keywords:** money market; general equilibrium; microfoundations; economic growth; inflation

**JEL Classification:** D01; D50; E40; E52

## 1. Introduction

The money market is an inseparable part of the structure of the economy. Accordingly, monetary policy is always associated with economic activities in which the role of money is emphasized in explaining economic fluctuations and inflation (Tavlas 1981). Therefore, the importance of the money market to the economy is the central topic in economic ideology and theory. The theory of money has developed with two contrasting views concerning the importance of money and monetary policy (Friedman 1956): the Fisherian view emphasizes controlling the quantity of money through the monetary policy, and the Keynesian view emphasizes aggregate demand through fiscal policy. The views on monetary instruments and policies that influence economic growth, inflation, and employment have been the subject of debate in theory and practice throughout the past century (Friedman 1968). Thus, the role of money and the transmission channel of monetary policy are rehabilitated (Adrian and Shin 2009).

The question in the monetary analysis debate is whether the quantity of money is exogenous (determined by money authorities) or endogenous (determined by economic activities). The Fisherian view relies on the classical quantity theory of money, in which monetarists argued that money is neutral in the long term and should also be in the short term. Changes in the general price level were linked to proportionate changes in the money supply (Fisher 1911). The Keynesian view relies on Keynes's macroeconomic theory, in which money has real effects both in the long term and in the short term (Gaffard 2018). Keynes's revolutionary contribution to economics lies in the theory of effective demand (Keynes 1930), in which money demand relies upon the national outcome determined by the IS-LM model (Hicks 1937; Hansen 1953). However, the IS-LM model was weak and unreliable due to a lack of microfoundations for general equilibrium. Meanwhile, the classical theory of general equilibrium (Walras 1874; Arrow and Debreu 1954) has not been conducted the money market in the general equilibrium models. According to Lucas (1972), the monetary system utilities the assumption of nominal price stickiness

and embodies the framework of general equilibrium. Therefore, the new neoclassical synthesis combined neoclassical economics and Keynesian macroeconomics to provide microfoundations for dynamic stochastic general equilibrium. However, the dynamic stochastic general equilibrium (DSGE) model still fails to incorporate critical aspects of economic behavior and the financial sector to predict or respond to a financial crisis (Blanchard 2016; Stiglitz 2018).

Although many monetary theories and models have been developed for monetary policy analysis in economic development, the underlying monetary theories are deficient with solid foundations in linking the money market with economic activities. For these reasons, this paper integrates the money market into the general equilibrium framework. There are specific objectives that are carried out in this research. First, the monetary theory is developed upon the quality theory of money, Keynesian macroeconomic theory, and classical theory of general equilibrium. Second, the research develops microfoundations that are the key to explaining the market behaviors and general equilibrium mechanism. Last, the linking mechanism between the money market and economic activities is conducted under three typical cases: full-employment equilibrium economy, steady-state equilibrium economy, and sticky-price equilibrium economy.

## 2. Literature Review

Classical monetary theory is well-known as the quantity theory of money (Fisher 1911). The quantity theory of money emphasizes the role of money supply, assuming that the velocity of money and the national outcome are constant. The quantity theory of money argues that increasing the quantity of money will increase the price with the same proportion. However, the classical theory of money has not addressed the tradeoff between inflation and national outcome. Pigou (1917) emphasized the Marshallian view on money demand and the Fisherian view on the money supply, which was formulated in terms of the velocity of money circulation. Nevertheless, there seems to be one basic proposition characterizing the quantity theory of money. The money supply exogenously engineered by the monetary authorities has the long-run effect on a change in the price level (and other nominal variables) of the same proportion as the money stock, but no change in any real variables (McCallum and Nelson 2010). Monetarists (Friedman and Schwartz 1965; Friedman 1968) argued that an increase in the quantity of money will increase real GDP in the short run, but that money is neutral in the long run. However, the quantity theory of money is criticized by the assumption of a constant velocity of money (Mishkin 2001) and the neutrality of money in the long run (Evans 1996).

The Keynesian monetary theory emphasizes the money demand stemming from economic activities and the role of fiscal policy in promoting economic growth. Liquidity preference theory combines money demand (liquidity preference) with the money supply by central banks to determine the equilibrium interest rate (Keynes 1936). An increase in the money supply that lowers the interest rate positively affects the marginal efficiency of capital. This will promote the expansion of economic activities and increase the national outcome. In addition, Keynes was skeptical about the effectiveness of monetary policy when the economy is in a liquidity trap and because of uncertainty in the financial markets (Twinoburyo and Odhiambo 2018). The IS-LM model (Hicks 1937; Hansen 1953) explains the general equilibrium mechanism between the commodity market and money market through the intersection of the investment-savings (IS) and liquidity preference—money supply (LM). However, the IS-LM model lacks the precision and realism to be a useful prescription tool for economic policy. In both classical and Keynesian monetary theories, the assumption of exogenous money supply has been challenged and discarded in subsequent and modern theories (Romer 2006).

In recent literature, the modern monetary theory attempts to deal with the monetary-fiscal conflict based on practical experience in monetary and fiscal policies. Compared with the mainstream monetary theory, the modern monetary theory has contrasting views in funding government spending, achieving full employment, setting interest rates, and

controlling inflation rates. According to modern monetary theory, the government can create money funded by the central bank for government spending in pursuit of socio-economic goals. Therefore, modern monetary policy is suitable for an economy with low inflation and high unemployment. In addition, the government can reduce government spending and raise taxes only if the economy has an inflationary problem. However, economic researchers believe that modern monetary theory's monetary and fiscal policies can lead to high inflation, even hyperinflation, when the economy is in full employment. Moreover, since the modern monetary theory does not explain the interest rate mechanism and the relationship of money supply and money demand with economic activities, the modern monetary theory lacks a theoretical foundation for monetary policies and fiscal policy (Palley 2015).

The classical monetary theory lays the groundwork for developing new classical models (Goodfriend and King 1997; Palley 2007), in which the monetary policy does not affect real GDP according to the rational expectations hypothesis and real business cycle theory. Meanwhile, new Keynesian models assume that prices are rigid to fiscal or monetary policy (Goodfriend and King 1997). Meanwhile, the new consensus model is the product of the new classical model (rational expectations and real business cycles) and the new Keynesian model (price stickiness). The new consensus model suggests that monetary policy focuses on short-run output stabilization and long-run price stability (Fontana and Palacio-Vera 2007). In fact, monetary theories have been challenged on the grounds of the velocity of money and the instability of the money demand function (White 2013). Moreover, the underlying monetary models are the absence of money, exchange rate roles, inadequate treatment of markets (financial, labor, capital markets) (Arestis and Sawyer 2008).

Recently, monetary policy has been addressed in the general equilibrium framework (Woodford 2005; Christiano et al. 2010) through the monetary policy rules (the Taylor rule, Friedman's k-percent rule, and other interest targeting rules). However, the rule-based approaches are only suitable under certain circumstances. The lack of a theoretical foundation has resulted in limitations in the monetary theories and models. Therefore, monetary policies and rules pursued for a long time with potential risks will lead to stagnation phenomena and economic crisis. Monetary theory built on solid microfoundations is the key to understanding the nature of the money market. The linkage of the money market within the general equilibrium framework is the basis for explaining the transmission mechanism and the role of monetary and fiscal policy on economic growth and stability.

## 3. General Equilibrium

The general equilibrium theory describes the interaction mechanism between economic actors (household, business, government) in the markets (commodity market, resource market). Therefore, the structure of the economy is the starting point for building a general equilibrium theory. Figure 1 illustrates the simple economic system in which the national economy assumes no taxes and subsidies, no international trade and investment with the rest of the world. In such an economy, the aggregate commodity demand (AD) presents the relationship between the average commodity price ($p_{AD}$) and the total quantity demanded ($Q$) in the economy. From the GDP formula (Trinh 2018), the aggregate commodity demand (AD) includes household spending ($C = p_C \times Q_C$), government spending ($G = p_G \times Q_G$), and enterprise investment ($I = p_I \times Q_I$) rewritten as follows:

$$p_{AD} \times Q = p_C \times Q_C + p_G \times Q_G + p_I \times Q_I \tag{1}$$

where $Q = Q_C + Q_G + Q_I$ is the total production quantity of the economy and $p_{AD} = \frac{p_C \times Q_C + p_G \times Q_G + p_I \times Q_I}{Q}$ is the average commodity price in the economy.

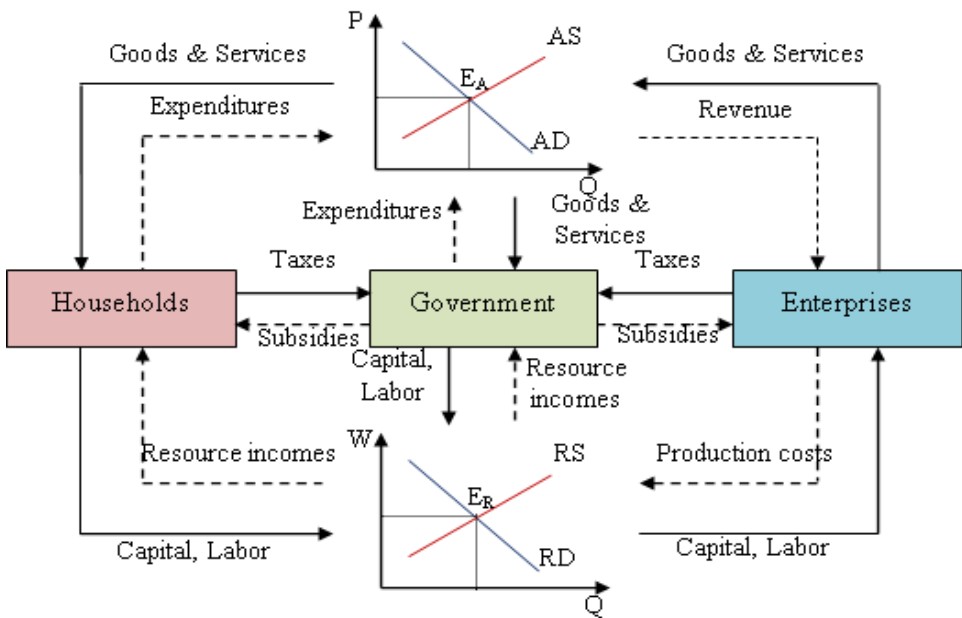

**Figure 1.** The structure of the simple economy.

The aggregate commodity supply (AS) is determined on the basis of the relationship between the market equilibrium and the marginal equilibrium in the commodity market. The aggregate commodity supply (AS) intersects the aggregate commodity demand (AD) at the equilibrium quantity ($Q_A$), where total marginal revenue ($MR_A$) is equal to total marginal cost ($MC_A$) in the commodity market (Trinh 2018, 2019) as illustrated in Figure 2. Based on this relationship, the aggregate commodity supply (AS) is determined as follows:

$$p_{AS} = MC_A - p'_{AD}(Q) \times Q \tag{2}$$

where $p'_{AD}(Q)$ is the first derivative of the aggregate commodity demand function $p'_{AD}(Q)$ $MC_A$ is determined upon the first derivative of the total cost function ($TC_A$) in the commodity market.

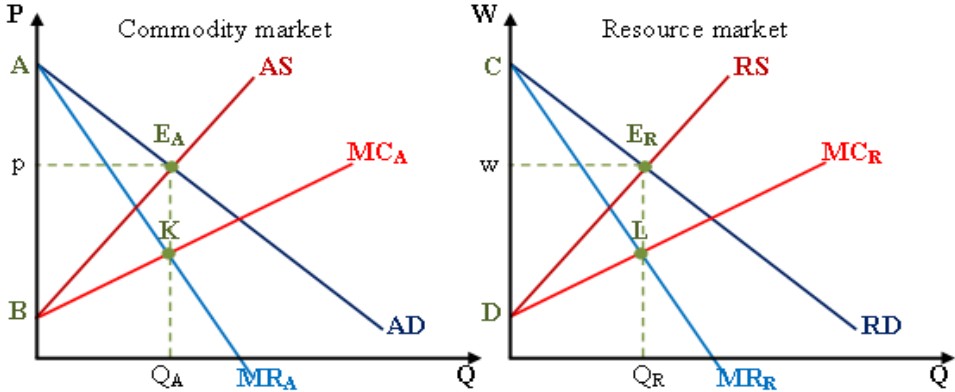

**Figure 2.** The equilibriums of AD-AS and RD-RS.

The relationship between total production cost ($TC_A$) in the commodity market and total revenue ($TR_R = w_{RD} \times Q$) in the resource market is illustrated as follows:

$$TC_A = w_K \times K + w_L \times L = w_{RD} \times Q = TR_R \tag{3}$$

$$MC_A = w'_{RD}(Q) \times Q + w_{RD} \tag{4}$$

where $w_{RD} = \frac{w_K \times K + w_L \times L}{Q}$ is the average resource price and $Q$ is the total production quantity in the economy.

Substituting $MC_A = w'_{RD}(Q) \times Q + w_{RD}$ into Equation (2), the aggregate commodity supply function is rewritten as follows:

$$p_{AS} = w_{RD} + \left(w'_{RD}(Q) - p'_{AD}(Q)\right) \times Q \tag{5}$$

The aggregate resource supply (RS) is determined by the relationship between the market equilibrium and the marginal equilibrium in the resource market. The market equilibrium of the resource at $Q_R$ where total marginal revenue ($MR_R$) is equal to the total marginal cost ($MC_R$) in the resource market as illustrated in Figure 2. At this equilibrium quantity, the aggregate resource supply (RS) will intersect the aggregate resource demand (RD) in the resource market. Based on this relationship, the aggregate resource supply (RS) is determined as follows:

$$w_{RS} = MC_R - w'_{RD}(Q) \times Q \tag{6}$$

where $w'_{RD}(Q)$ is the first derivative of the aggregate resource demand function $w_{RD}(Q)$ $MC_R$ is determined on the first derivative of the total cost function ($TC_R$) in the resource market. The relationship between the total cost ($TC_R$) in the resource market and the total revenue ($TR_A = p_{AD} \times Q$) in the commodity market is illustrated as follows:

$$TC_R = p_{AD} \times Q = TR_A \tag{7}$$

$$MC_R = p'_{AD}(Q) \times Q + p_{AD} \tag{8}$$

Replacing $MC_R = p'_{AD}(Q) \times Q + p_{AD}$ into Equation (6), the aggregate resource supply function (RS) is rewritten as follows:

$$w_{RS} = p_{AD} + \left(p'_{AD}(Q) - w'_{RD}(Q)\right) \times Q \tag{9}$$

The aggregate supply is identified from the relation between market and marginal equilibrium, relying on its aggregate demand and marginal cost (Trinh 2020, 2021). From this base, the total economic surplus in the commodity market is the area AKB (equals the area $AE_AB$). Similarly, the total economic surplus in the resource market is the area CLD (equals the area $CE_RD$), as illustrated in Figure 2.

The general equilibrium of the economy occurs only when the equilibrium quantity of the commodity market ($Q_A$) equals the equilibrium quantity of the resource market ($Q_R$) as illustrated in Figure 3. When the equilibrium quantity of the commodity market ($Q_A$) is greater than (or less than) the equilibrium quantity of the resource market ($Q_R$), the economy is in general disequilibrium status. The concept of the economic surplus is the crucial basis of behavior analysis and decision-making of economic actors in the markets. While the market equilibrium represents each economic actor's partial (individual) rational choice, the general equilibrium represents economic actors' general (cooperative) rational choice. When the economy reaches general equilibrium at the equilibrium quantity ($Q_E$), the average commodity price is determined at $p_E$, and the average resource cost is determined at $w_E$, as illustrated in Figure 3. Since the total economic surplus (economic welfare) is the sum of AKB and CLD, the economic welfare (total economic surplus in the commodity and resource markets) is maximized in the general equilibrium economy.

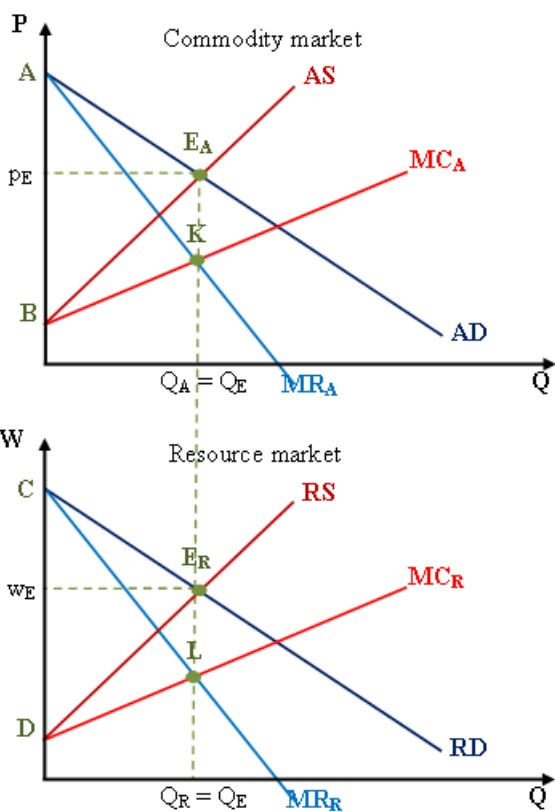

**Figure 3.** General equilibrium of the economy.

## 4. Money Market

The money market presents the relationship between the demand and supply of money. The demand for money represents the relationship between the nominal interest rate (the price of money) and the quantity demanded of money over a given period. The money supply presents the relationship between the nominal interest rate (the price of money) and the quantity supplied of money over a given period. The equilibrium of the money market occurs at the nominal interest rate (equilibrium price of money), where the quantity demanded of money is equal to the quantity supplied of money, as illustrated in Figure 4.

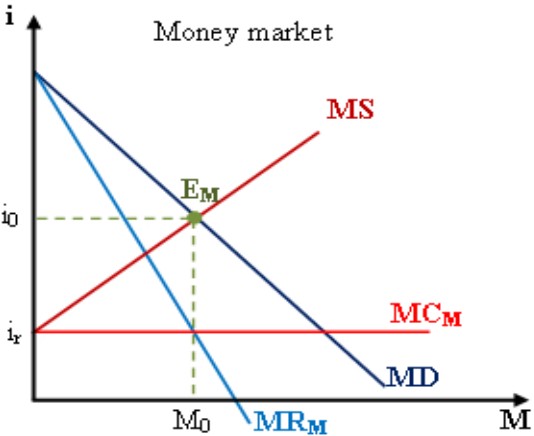

**Figure 4.** The equilibrium of the money market.

Total revenue ($TR_M$) and marginal revenue ($MR_M$) on the money market are determined as follows:

$$TR_M = i_D \times M \tag{10}$$

$$MR_M = i'_D \times M + i_D \tag{11}$$

where $i_D$ is the money demand function and $i'_D$ is the first derivative of the money demand function.

Since the nominal interest rate ($i_D$) is defined as the point where the money supply intersects the money demand, the real interest rate ($i_r$) is defined as the marginal cost of money ($MC_M$). This definition is relevant to Fisher's monetary theory (Fisher 1911, 1930) and empirical research (Fama 1975), in which the real interest rate is constant through time and approximately equal to the nominal interest rate minus the expected inflation rate.

$$MC_M = i_r \tag{12}$$

Figure 4 illustrates the relationship between market equilibrium and marginal equilibrium in the money market. At the equilibrium money quantity ($M_E$), where the money demand curve intersects the money supply curve ($i_D = i_S$), the marginal revenue of money ($MR_M$) is also equal to the marginal cost of money ($MC_M$).

From Equations (11) and (12), the money supply function is determined at $MR_M = MC_M$ and replacing $i_D$ by $i_D$ as follows:

$$i_S = i_r - i'_{\backslash D} \times M \tag{13}$$

From Equation (13), the money supply function ($i_S$) relies on the money demand ($i_D$) and marginal cost of money ($i_r$).

According to the standard Fisher hypothesis (Fisher 1930), the nominal interest rate ($i$) relies on the real interest rate ($i_r$) and the inflation rate ($f$) as in Equation (14), in which the real interest rate is estimated as return on treasury bills that seems to be constant through time. Meanwhile, the inverted Fisher hypothesis (Carmichael and Stebbing 1983) states the constancy of the nominal interest rate relative to inflation and the inverse correlation between inflation and the real interest rate.

According to the Fisherian approach, the nominal interest rate ($i$) is defined as the real interest rate ($i_r$) plus the inflation rate ($f$).

$$i = i_r + f \tag{14}$$

From Equations (13) and (14), the relationship between inflation ($f$) and the money demand ($i_D$) and the money quantity supplied ($M$) is as follows:

$$f = -i'_{\backslash D} \times M \tag{15}$$

The quantity theory of money (Fisher 1911) expresses the relationship between the quantity of money ($M$) and the nominal national income ($Y$) as follows:

$$M \times V = Y = p \times Q \tag{16}$$

where $V$ is the velocity of money. The nominal national income ($Y$) is determined by the price ($p$) and quantity ($Q$) when the economy is in the general equilibrium status. Figure 5 illustrates the relationship between the money market and the commodity market in general equilibrium status. From Equation (16), the formula of money quantity ($M$) is computed as follows:

$$M = \frac{p \times Q}{V} \tag{17}$$

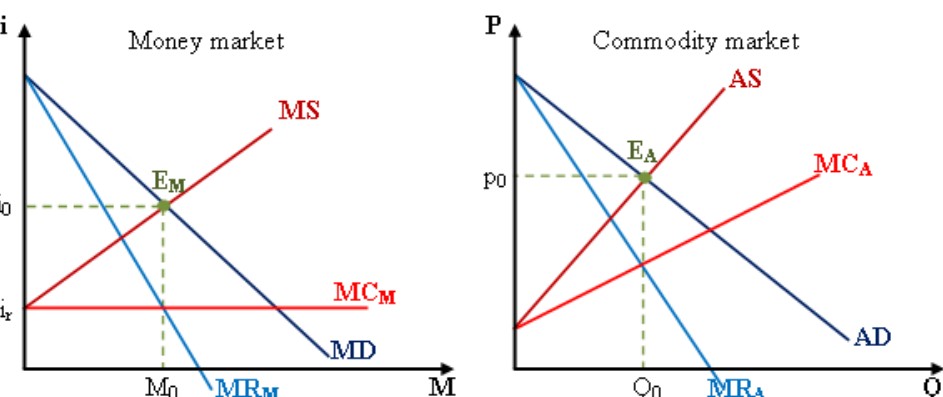

**Figure 5.** Money market and general equilibrium.

Equation (17) reveals that the money supply (*M*) depends on the nominal national income (*Y* = *p* × *Q*) and the velocity of money (*V*). The nominal national income (*Y*) derives from investments in both real assets and financial assets. Also, the velocity of money (the money turnover) must be estimated upon economic activities associated with the real and financial assets.

In the case of the full-employment equilibrium economy, an increase in the money supplied (*M*) will increase the nominal interest rate (*i*) and inflation (*f*), respectively. From the formulation of money quantity (17), when *Q* and *V* are constant, the growth rate of money (*g*) in the money market will correspond to the growth rate of the price level in the commodity market, as illustrated in Figure 6.

$$M_1 = \frac{p \times (1+g) \times Q}{V} = M \times (1+g) \tag{18}$$

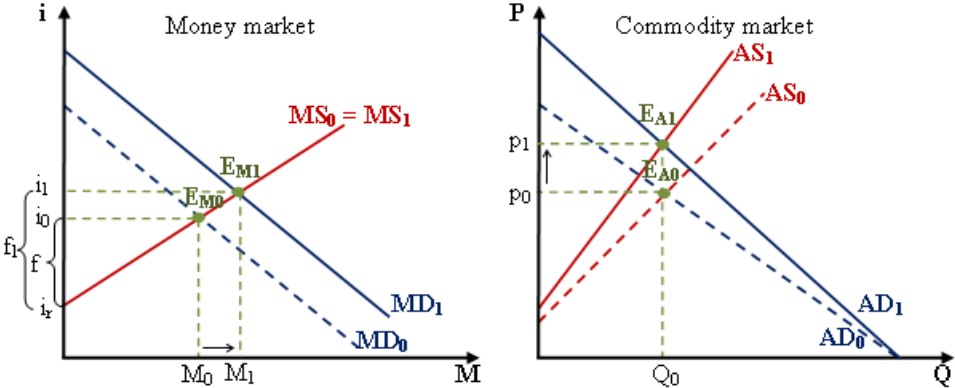

**Figure 6.** The full-employment equilibrium economy.

Equation (19) is developed upon the standard Fisher hypothesis. However, the inverted Fisher hypothesis may be concerned as the structure of the money market is extended with the bond market and regulations on interest rates.

$$i_1 = i_r + f_1 = i_r + f \times (1+g) \tag{19}$$

In such a case, the price level in the economy and inflation in the money market have the same growth rates. This growth rate leads to an increase in the nominal GDP but has no impact on economic activity and the economy's structure. This full-employment monetary model is relevant to the classical theory of money in explaining that monetary policies do not affect the real GDP. Thus, the monetary policies focus on money demand as the main driving force that affects the interest rate in the money market.

In the case of the steady-state equilibrium economy, the economic growth responds with the market scale in which there are increases in the quantity ($Q$) and the price ($p$) at the same rate as shown in Figure 7. Then, the money supplied ($M_1$) in the case of the steady-state equilibrium is determined as follows:

$$M_1 = \frac{p \times (1+g) \times Q \times (1+g)}{V} = M \times (1+g)^2 \tag{20}$$

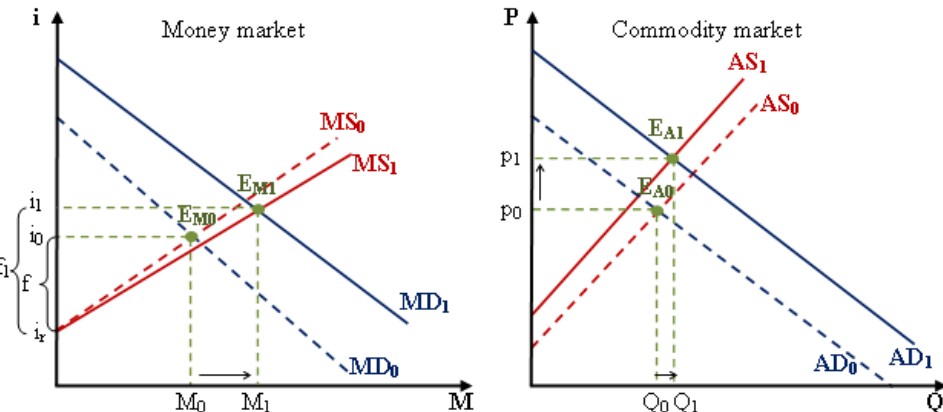

**Figure 7.** The steady-state equilibrium economy.

From Equation (20), the monetary model may extend with the real business cycle hypothesis, in which the levels of price and quantity increase with different growth rates, but the price level in the economy and inflation in the money market are always the same growth rate as in Equation (21).

Equation (20) may be extended and applied for the real economy under economic policies (or economic shocks) in which the growth rate of the quantity and price are not the same. However, the growth rates of inflation and the price level are always the same as in Equation (21).

$$i_1 = i_r + f_1 = i_r + f \times (1+g) \tag{21}$$

When the equilibrium quantity increases from $Q \rightarrow Q_1$ and the equilibrium price increases from $p \rightarrow p_1$, the money quantity increases from $M \rightarrow M_1$. The aggregate demand and aggregate supply increase at the same rate ($g$) to reach the steady-state equilibrium status. Assuming the real interest rate (or marginal cost of money) remains constant, the money supply function ($i_S$) and the money demand function ($i_D$) are illustrated in Figure 7.

The steady-state monetary model is relevant with the classical theory of general equilibrium and the new classical model with rational expectations hypothesis, in which the levels of price and quantity increase with the same growth rate as the nominal GDP. The monetary policy focuses on both monetary demand and money supply in which the money supply responds to changes in nominal interest rate.

In the case of the sticky-price equilibrium economy, the economic growth (nominal national income) is due to an increase in the quantity ($Q_1 = Q \times (1 + g)$) and no increase in the price. Then, the money supplied ($M_1$) increases with the growth rate of the equilibrium quantity of the economy as follows:

$$M_1 = \frac{p \times Q \times (1+g)}{V} = M \times (1+g) \tag{22}$$

$$i_1 = i_r + f_1 = i_r + f \tag{23}$$

When the equilibrium quantity increases from $Q \rightarrow Q_1$ and the equilibrium price remains constant $p = p_1$, the money quantity increases from $M \rightarrow M_1$ while the nominal

interest rate ($i$) and inflation rate ($f$) remain constant for the general equilibrium economy under the sticky price. The demand and supply of money are illustrated in Figure 8.

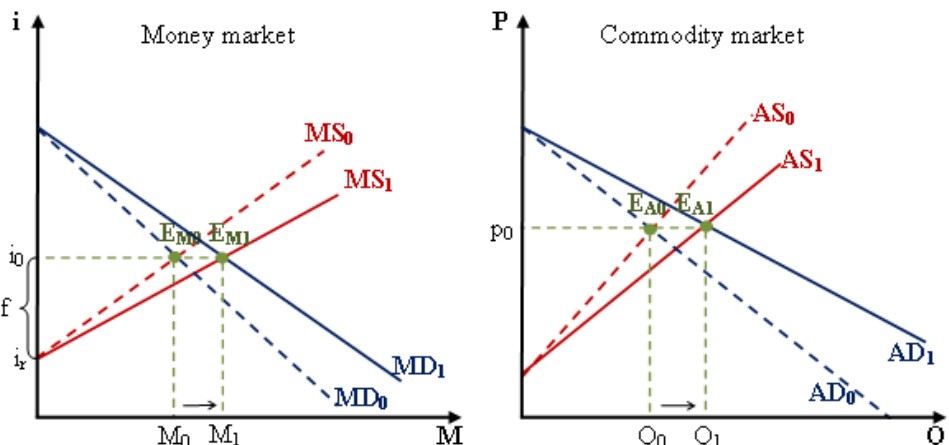

**Figure 8.** The sticky-price equilibrium economy.

The sticky-price monetary model is relevant with the Keynesian monetary theory and the new Keynesian models with nominal rigidity hypothesis, in which the price and wage are rigid to changes in fiscal and monetary policies. Thus, the sticky-price monetary policies focus on money demand and money supply to maintain a stable interest rate.

These monetary models are the extreme of monetary policies in the real economy that provide the key implications for monetary instruments and policies in the real economy. Combining these monetary models also provides a theoretical foundation for the modern monetary theory and the new consensus models.

## 5. Numerical Example

This section illustrates numerical examples of the relationship between the money market and the economy's structure. The quantity theory of money and the Fisher equation determine the money demand and money supply functions based on the underlying assumptions. Assuming an economy in which the aggregate commodity demand function ($p_{D0}$) and the aggregate resource demand function ($w_{D0}$) are given as follows:

$$p_{D0} = 40 - 0.2Q \tag{24}$$

$$w_{D0} = 30 - 0.1Q \tag{25}$$

The aggregate commodity supply function ($p_{S0}$) and the aggregate resource supply function ($w_{S0}$) are determined as follows:

$$p_{S0} = w_{D0} + \left(w'_{D0} - p'_{D0}\right) \times Q = 30 - 0.1Q + (-0.1 + 0.2)Q = 30 \tag{26}$$

$$w_{S0} = p_{D0} + \left(p'_{D0} - w'_{D0}\right) \times Q = 40 - 0.2Q + (-0.2 + 0.1)Q = 40 - 0.3Q \tag{27}$$

The commodity market equilibrium at the point $E_{A0}(p_0, Q_{A0})$ and the resource market equilibrium at the point $E_{R0}(w_0, Q_{R0})$ are computed as follows:

$$Q_{A0} : p_{D0} = p_{S0} \Rightarrow 40 - 0.2Q_{A0} = 30 \Rightarrow Q_{A0} = 50; p_0 = 30 \tag{28}$$

$$Q_{R0} : w_{D0} = w_{S0} \Rightarrow 30 - 0.1Q_{R0} = 40 - 0.3Q_{R0} \Rightarrow Q_{R0} = 50; w_0 = 25 \tag{29}$$

Since $Q_{A0} = Q_{R0} = Q_0$, the economy is in the general equilibrium status at the equilibrium quantity ($Q_0 = 50$), as illustrated in Figure 9.

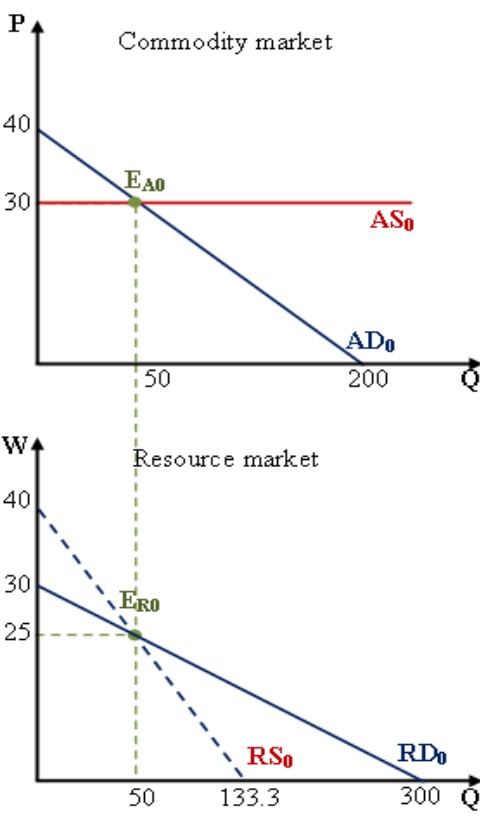

**Figure 9.** General equilibrium of the hypothetical economy.

Assume the economy has the current money quantity ($M_0 = 300$), the current nominal interest rates $i_0 = 7\%$, and inflation rate $f_0 = 5\%$. According to the Fisher equation, the real interest rate ($i_r$) is determined as follows:

$$i_r = i_0 - f_0 = 7\% - 5\% = 2\% \tag{30}$$

From the formula of money quantity, the velocity of money ($V$) is computed as follows:

$$V = \frac{p_0 \times Q_0}{M_0} = \frac{30 \times 50}{300} = 5 \tag{31}$$

Supposing that the money supply function ($i_S$) and the money demand function ($i_D$) are linear and intersected at the equilibrium point $E_{M0}(i_0 = 7\%, M_0 = 300)$ as follows:

$$i_S = i_r - i'_D \times M \Rightarrow 7\% = 2\% - i'_D \times 300 \Rightarrow i'_D = -\frac{5\%}{300} \tag{32}$$

$$i_D = a + i'_D \times M \Rightarrow 7\% = a - \frac{5\%}{300} \times 300 \Rightarrow a = 12\% \tag{33}$$

The money supply function ($i_S$) and the money demand function ($i_D$) are rewritten as below.

$$i_S = 2\% + \frac{5\%}{300} \times M \tag{34}$$

$$i_D = 12\% - \frac{5\%}{300} \times M \tag{35}$$

Figure 10 illustrates the market equilibrium of money for the above example.

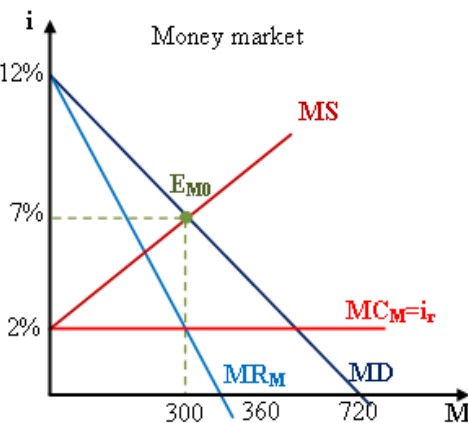

**Figure 10.** Money market equilibrium of the hypothetical economy.

The monetary models toward the money market in the general equilibrium framework are considered for three cases: (1) the full-employment equilibrium economy; (2) the steady-state equilibrium economy; and (3) the sticky-price equilibrium economy.

For the full-employment equilibrium economy, the economic growth (nominal national income) is due to an increase in the equilibrium price level $p \to p_1$ while the equilibrium quantity ($Q$) remains constant. Assuming that the equilibrium price level rises at a growth rate of $g = 10\%$, then the aggregate commodity demand function ($p_{D1}$) and the aggregate resource demand function ($w_{D1}$) are determined as follows:

$$p_{D1} = p_{D0} \times (1 + g) = (40 - 0.2Q) \times (1 + 0.1) = 44 - 0.22Q \tag{36}$$

$$w_{D1} = w_{D0} \times (1 + g) = (30 - 0.1Q) \times (1 + 0.1) = 33 - 0.11Q \tag{37}$$

The aggregate commodity supply function ($p_{S1}$) and the aggregate resource supply function ($w_{S1}$) are determined as follows:

$$p_{S1} = w_{D1} + (w'_{D1} - p'_{D1}) \times Q = 33 - 0.11Q + (-0.11 + 0.22)Q = 33 \tag{38}$$

$$w_{S1} = p_{D1} + (p'_{D1} - w'_{D1}) \times Q = 44 - 0.22Q + (-0.22 + 0.11)Q = 44 - 0.33Q \tag{39}$$

The commodity market equilibrium at the point $E_{A1}(p_1, Q_{A1})$ and the resource market equilibrium at the point $E_{R1}(w_1, Q_{R1})$ are computed as follows:

$$Q_{A1} : p_{D1} = p_{S1} \Rightarrow 44 - 0.22Q_{A1} = 33 \Rightarrow Q_{A1} = 50; p_1 = 33 \tag{40}$$

$$Q_{R1} : w_{D1} = w_{S1} \Rightarrow 33 - 0.11Q_{R1} = 44 - 0.33Q_{R1} \Rightarrow Q_{R1} = 50; w_1 = 27.5 \tag{41}$$

Since $Q_{A1} = Q_{R1} = 50$, the economy is in general equilibrium status with the equilibrium quantity $Q_1 = 50$ and the equilibrium price $p_1 = 33$. According to the quantity theory of money, the money quantity supplied ($M_1$) that assumes a constant velocity of money ($V = 5$) depends on the quantity $Q_1$ and the price level $p_1$ as follows:

$$M_1 = \frac{p_1 \times Q_1}{V} = \frac{33 \times 50}{5} = 330 \tag{42}$$

Since the growth rate of inflation reflects the growth rate of the price level ($p_1 = p_0 \times (1 + g)$), the inflation rate ($f_1$) is determined as follows:

$$f_1 = f \times (1 + g) = 5\% \times (1 + 0.1) = 5.5\% \tag{43}$$

According to the Fisher equation, the nominal interest rate ($i_1$) is computed as follows:

$$i_1 = i_r + f_1 = 2\% + 5.5\% = 7.5\% \tag{44}$$

Since the money supply function ($i_{S1}$) and money demand function ($i_{D1}$) intersect at the equilibrium point $E_{M1}(i_1 = 7.5\%, M_1 = 330)$, the money supply function ($i_{S1}$) and the money demand function ($i_{D1}$) are determined as follows:

$$i_{S1} = i_r - i'_{D1} \times M_1 \Rightarrow 7.5\% = 2\% - i'_{D1} \times 330 \Rightarrow i'_{D1} = -\frac{5.5\%}{330} \tag{45}$$

$$i_{S1} = 2\% + \frac{5.5\%}{330} \times M \tag{46}$$

$$i_{D1} = a + i'_{D1} \times M \Rightarrow a = 7.5\% + \frac{5.5\%}{330} \times 330 = 13\% \tag{47}$$

$$i_{D1} = 13\% - \frac{5.5\%}{330} \times M \tag{48}$$

When the economy is in the full-employment equilibrium status, the money market equilibrium and the general equilibrium of the economy are illustrated in Figure 11.

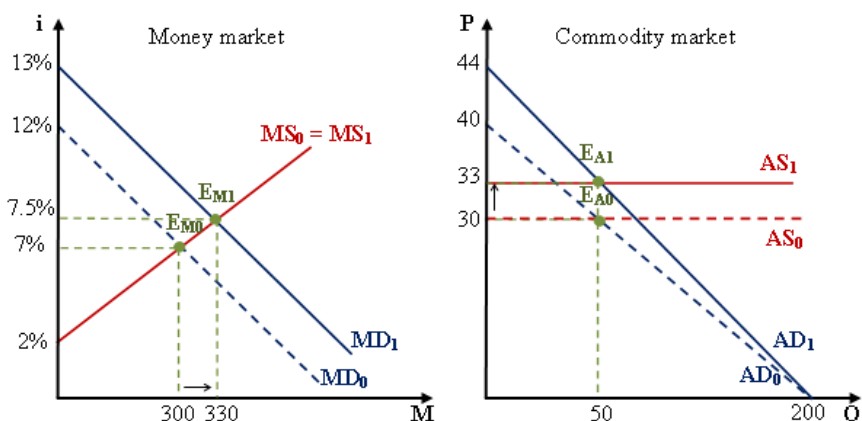

**Figure 11.** Case of the full-employment equilibrium economy.

For the steady-state equilibrium economy, the economic growth (nominal national income) is due to an increase in the equilibrium price level $p \rightarrow p_1$ and an increase in the equilibrium quantity $Q \rightarrow Q_1$ at the same rate assuming $g = 10\%$. Then, the aggregate commodity demand function ($p_{D1}$) is determined as follows:

$$p_{D1} = p_{D0} \times (1 + g) = (40 - 0.2Q_{D0}) \times (1 + 0.1) = 44 - 0.22Q_{D0} \tag{49}$$

$$\Rightarrow Q_{D0} = 200 - \frac{1}{0.22} \times p_{D1} \tag{50}$$

$$Q_{D1} = Q_{D0} \times (1 + g) = \left(200 - \frac{1}{0.22} \times p_{D1}\right) \times (1 + 0.1) \tag{51}$$

$$\Rightarrow p_{D1} = 44 - 0.2Q \tag{52}$$

Similarly, the aggregate resource demand function ($w_{D1}$) is determined as follows:

$$w_{D1} = w_{D0} \times (1 + g) = (30 - 0.1Q_{D0}) \times (1 + 0.1) = 33 - 0.11Q_{D0} \tag{53}$$

$$\Rightarrow Q_{D0} = 300 - \frac{1}{0.11} \times w_{D1} \tag{54}$$

$$Q_{D1} = Q_{D0} \times (1 + g) = \left(300 - \frac{1}{0.11} \times w_{D1}\right) \times (1 + 0.1) \tag{55}$$

$$\Rightarrow w_{D1} = 33 - 0.1Q \tag{56}$$

The aggregate commodity supply function ($p_{S1}$) and the aggregate resource supply function ($w_{S1}$) are determined as follows:

$$p_{S1} = w_{D1} + \left(w'_{D1} - p'_{D1}\right) \times Q = 33 - 0.1Q + (-0.1 + 0.2)Q = 33 \tag{57}$$

$$w_{S1} = p_{D1} + \left(p'_{D1} - w'_{D1}\right) \times Q = 44 - 0.2Q + (-0.2 + 0.1)Q = 44 - 0.3Q \tag{58}$$

The commodity market equilibrium at the point $E_{A1}(p_1, Q_{A1})$ and the resource market equilibrium at the point $E_{R1}(w_1, Q_{R1})$ are computed as follows:

$$Q_{A1} : p_{D1} = p_{S1} \Rightarrow 44 - 0.2Q_{A1} = 33 \Rightarrow Q_{A1} = 55; p_1 = 33 \tag{59}$$

$$Q_{R1} : w_{D1} = w_{S1} \Rightarrow 33 - 0.1Q_{R1} = 44 - 0.3Q_{R1} \Rightarrow Q_{R1} = 55; w_1 = 27.5 \tag{60}$$

Since $Q_{A1} = Q_{R1} = 55$, the economy is in general equilibrium status with the equilibrium quantity $Q_1$ = 55 and the equilibrium price $p_1$ = 33. According to the quantity theory of money, the money quantity supplied ($M_1$) that assumes a constant velocity of money ($V$ = 5) depends on the quantity $Q_1$ and the price level $p_1$ as follows:

$$M_1 = \frac{p_1 \times Q_1}{V} = \frac{33 \times 55}{5} = 363 \tag{61}$$

Since the growth rate of inflation reflects the growth rate of the price level ($p_1 = p_0 \times (1 + g)$), the inflation rate ($f_1$) is determined as follows:

$$f_1 = f \times (1 + g) = 5\% \times (1 + 0.1) = 5.5\% \tag{62}$$

According to the Fisher equation, the nominal interest rate ($i_1$) is computed as follows:

$$i_1 = i_r + f_1 = 2\% + 5.5\% = 7.5\% \tag{63}$$

Since the money supply function ($i_{S1}$) and money demand function ($i_{D1}$) intersect at the equilibrium point $E_{M1}(i_1 = 7.5\%, M_1 = 363)$ as illustrated in Figure 12, thus the money supply function ($i_{S1}$) and the money demand function ($i_{D1}$) are determined as follows:

$$i_{S1} = i_r - i'_{D1} \times M_1 \Rightarrow 7.5\% = 2\% - i'_{D1} \times 363 \Rightarrow i'_{D1} = -\frac{5.5\%}{363} \tag{64}$$

$$i_{S1} = 2\% + \frac{5.5\%}{363} \times M \tag{65}$$

$$i_{D1} = a + i'_{D1} \times M \Rightarrow a = 7.5\% + \frac{5.5\%}{363} \times 363 = 13\% \tag{66}$$

$$i_{D1} = 13\% - \frac{5.5\%}{363} \times M \tag{67}$$

For the sticky-price equilibrium economy, the economic growth (nominal national income) is due to an increase in the equilibrium quantity $Q \to Q_1$ at the growth rate assuming $g$ = 10% while the equilibrium price level ($p$) remains constant. Then, the aggregate commodity demand function ($p_{D1}$) is determined as follows:

$$Q_{D1} = Q_{D0} \times (1 + g) = \left(200 - \frac{1}{0.2} \times p_{D1}\right) \times (1 + 0.1) \tag{68}$$

$$\Rightarrow p_{D1} = 40 - \frac{0.2}{1.1}Q \tag{69}$$

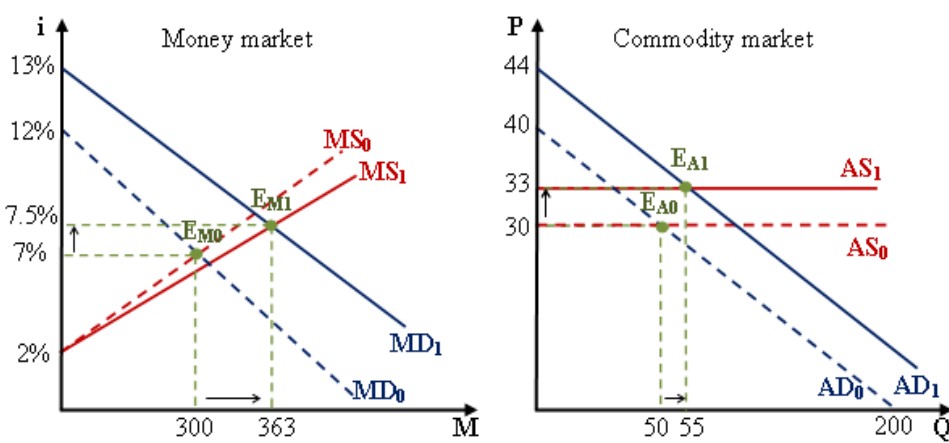

**Figure 12.** Case of the steady-state equilibrium economy.

Similarly, the aggregate resource demand function ($w_{D1}$) is determined as follows:

$$Q_{D1} = Q_{D0} \times (1+g) = \left(300 - \frac{1}{0.1} \times w_{D1}\right) \times (1+0.1) \tag{70}$$

$$\Rightarrow w_{D1} = 30 - \frac{0.1}{1.1}Q \tag{71}$$

The aggregate commodity supply function ($p_{S1}$) and the aggregate resource supply function ($w_{S1}$) are determined as follows:

$$p_{S1} = w_{D1} + \left(w'_{D1} - p'_{D1}\right) \times Q = 30 - \frac{0.1}{1.1}Q + \left(-\frac{0.1}{1.1} + \frac{0.2}{1.1}\right)Q = 30 \tag{72}$$

$$w_{S1} = p_{D1} + \left(p'_{D1} - w'_{D1}\right) \times Q = 40 - \frac{0.2}{1.1}Q + \left(-\frac{0.2}{1.1} + \frac{0.1}{1.1}\right)Q = 40 - \frac{0.3}{1.1}Q \tag{73}$$

The commodity market equilibrium at the point $E_{A1}(p_1, Q_{A1})$ and the resource market equilibrium at the point $E_{R1}(w_1, Q_{R1})$ are computed as follows:

$$Q_{A1} : p_{D1} = p_{S1} \Rightarrow 40 - \frac{0.2}{1.1}Q_{A1} = 30 \Rightarrow Q_{A1} = 55; p_1 = 30 \tag{74}$$

$$Q_{R1} : w_{D1} = w_{S1} \Rightarrow 30 - \frac{0.1}{1.1}Q_{R1} = 40 - \frac{0.3}{1.1}Q_{R1} \Rightarrow Q_{R1} = 55; w_1 = 25 \tag{75}$$

Since $Q_{A1} = Q_{R1} = 55$, the economy is in general equilibrium status with the equilibrium quantity $Q_1 = 55$ and the equilibrium price $p_1 = 30$. According to the quantity theory of money, the money quantity supplied ($M_1$) that assumes a constant velocity of money ($V = 5$) depends on the quantity $Q_1$ and the price level $p_1$ as follows:

$$M_1 = \frac{p_1 \times Q_1}{V} = \frac{30 \times 55}{5} = 330 \tag{76}$$

Since the equilibrium price remains constant, the growth rate of inflation will be zero $g = 0$, the inflation rate ($f_1$) is given as follows:

$$f_1 = f \times (1+g) = 5\% \times (1+0) = 5\% \tag{77}$$

According to the Fisher equation, the nominal interest rate ($i_1$) is computed as follows:

$$i_1 = i_r + f_1 = 2\% + 5\% = 7\% \tag{78}$$

Since the money supply function ($i_{S1}$) and money demand function ($i_{D1}$) intersect at the equilibrium point $E_{M1}(i_1 = 7.5\%, M_1 = 330)$, thus the money supply function ($i_{S1}$) and the money demand function ($i_{D1}$) are determined as follows:

$$i_{S1} = i_r - i'_{D1} \times M_1 \Rightarrow 7\% = 2\% - i'_{D1} \times 330 \Rightarrow i'_{D1} = -\frac{5\%}{330} \tag{79}$$

$$i_{S1} = 2\% + \frac{5\%}{330} \times M \tag{80}$$

$$i_{D1} = a + i'_{D1} \times M \Rightarrow a = 7\% + \frac{5\%}{330} \times 330 = 12\% \tag{81}$$

$$i_{D1} = 12\% - \frac{5\%}{330} \times M \tag{82}$$

When the economy is in the sticky-price equilibrium status, the money market equilibrium and the general equilibrium of the economy are illustrated as in Figure 13.

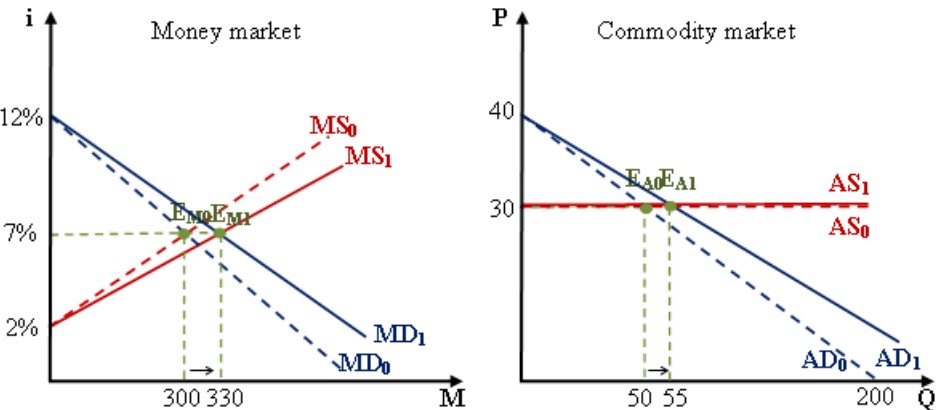

**Figure 13.** Case of the sticky-price equilibrium economy.

## 6. Conclusions

The monetary theory is developed upon microfoundations to understand the nature and behavior of the money market. The microfoundations are also an important basis for the explanation of the interrelationship between the commodity market and the resource market in the economy. Keynesian macroeconomic theory and classical general equilibrium theory are approached to define aggregate demand-aggregate supply and explain the general equilibrium mechanism. From these microfoundations together with the quantity theory of money and the Fisher equation, the monetary theory explains money demand—money supply, the relationship between interest rate and inflation through the equilibrium mechanism in the money market.

The monetary models illustrate the transmission mechanism between the money market and the economy under three cases: when the economy is in full-employment equilibrium status, an increase in the money supply increases the nominal price level without affecting actual economic productivity and the real price level. As a result, increasing the money supply raises price and inflation in the same proportions. If the economy is in under-employment equilibrium status, an increase in the money supply will stimulate economic activities, leading to aggregate demand and aggregate supply changes. While inflation reflects the rise in the price level, the increase in money supply relies on the national income (increases in the price and the quantity of the commodity market). When the economy is in sticky-price equilibrium status, an increase in the money supply increases economic activity in the direction of increasing the quantity level without changes in the price level (or unchanged inflation). Since the inflation and interest rates do not change in the money market, the money supply will change in the same proportion as the national income.

The paper contributes to the development of the monetary theory from inheriting the underlying monetary theories with refining the rigorous microfoundations. The monetary models provide the solid foundation for both theoretical and empirical researches on the role of fiscal and monetary policies in economic growth and macroeconomic stability. However, this research still has several limitations, which are also suggestions for further research related to: the nature of the velocity of money, the expansion of money market structure, the uncertainty of monetary demand, and the commodity market structure (aggregate demand-aggregate supply) under economic policies or economic shocks.

**Funding:** This research is funded by the Ministry of Education and Training (Vietnam) under project number B2020-DNA-12.

**Conflicts of Interest:** The author declares no conflict of interest regarding the publication of this paper.

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
