# Peer review of "Towards Money Market in General Equilibrium Framework"

_ijfs, doi:10.3390/ijfs10010012_

Round 1

Reviewer 1 Report

Accepted manuscript

Author Response

The author would express my sincere thanks for your revision and comments. The revised version of the paper has been improved upon your comments as follows:

- The Section 1 (Introduction) has been written to clarify the research gap and the objective of the research.

- This paper adds the new section of Literature review (Section 2) that provide much more literature in the underlying monetary theories and models dealing with this topic.

- This paper updates discussions in Section 4 (Money market) to highlight the relevance with and contribution to the literature.

- This paper has been checked grammars and spellings to make the readable paper (track change tool shows detail changes in the new version).

Reviewer 2 Report

The presented article is quite interesting paper dealing with good scientific topic. Its not very easy for the non-expert readers, since its full of economic and mathematical graphs and diagrams. On the other hand, its very interesting topic nowadays for the expert readers. 
Author presents methodological overview at the beginning of the article and all the scientific methods are appropriately used in the article. The scientific soundness of the topic mentioned in the article is good enough. The content of the article is good match with the topic of the journal, so the are no objections in this way. The citation and the list of references could be improved, because there is much more literature dealing with this topic, but its not major objection, which could cause the rejection of the article.
Overall merit is higher than average and could be published.

Author Response

(The authors gave the same response as above.)

Reviewer 3 Report

Review of the Manuscript  ID -ijfs-1556058 Towards Money Market in General Equilibrium Framework for the Journal of Risk and Financial Management.

General Comments

From my point of view, it is a very interesting topic and simultaneously it seems that to the best of my knowledge is the first empirical research which aims to integrate the money market into the structure of the economy. The paper consists of following sections: Introduction, General Equilibrium Framework, Money Market, Numerical Example and Conclusion.

However, I find some recommendations:

  1. The abstract must contain the main purpose of the paper, the research method used in the research and the main contributions.
  2. It would be very useful to add in the "Introduction" section the purpose, objectives and hypothesis of the research.
  3. We consider that the introduction should specify the novelty of the paper compared to other papers published in this area.
  4. Also, we consider the literature is not enough and that is why, we recommend the authors to refer to other recent works indexed in Web of Science, Scopus, Emerald, Cambridge, and of course MDPI Journals. We consider that the subject of the paper is also related to the tax compliance and that is why we recommend citing papers in journals such as:
  5. Nichita, A., Batrancea, l., Pop, C.M., Batrancea,I., Morar, I.D., Masca, E., Cesar, A.M.R., Forte, D.,  Formigoni, H.,  da Silva, A.A. (2019). We learn not for school but for life: Empirical evidence of the impact of tax literacy on tax compliance. Eastern European Economics 57: 397–429.
  6. Batrancea, L., Rathnaswamy Malar, M., Batrancea, I., Nichita, A., Rus, M.I, Tulai, H., Fatacean, G, Masca, E.S., Morar, I. D. (2020) Adjusted Net Savings of CEE and Baltic Nations in the Context of Sustainable Economic Growth: A Panel Data Analysis. J. Risk Financial Manag.13, 234.
  7. It should be interesting that the authors estimate the marginal effects before and after the financial crises to compare before and after this particular structural break.
  8. I propose to the authors the following structure of the paper: Introduction; Literature review; Methods and results; Discussions and conclusions.
  9. The Conclusions (not Conclusion) must be extended.

Author Response

The author would express my sincere thanks for your revision and comments. The revised version of the paper has been improved upon your comments as follows:

- The Section 1 (Introduction) has been written to clarify the research gap and the objective of the research.

- This paper adds the new section of Literature review (Section 2) that provide much more literature in the underlying monetary theories and models dealing with this topic.

- This paper updates discussions in Section 4 (Money market) to highlight the relevance with and contribution to the literature.

- This research style is the theoretical paper. Thus, the structure of the paper has some differences with the empirical paper. However, the paper has a numerical example as the theoretical illustration on the linking mechanism between money market with economic activities in general equilibrium framework. This theoretical research provides solid foundation for both theoretical and empirical researches related to monetary and fiscal policies on economic growth and stability.

- This paper has been checked grammars and spellings to make the readable paper (track change tool shows detail changes in the new version).